# Guiding Online Reinforcement Learning with Action-Free Offline Pretraining

## Abstract

Offline RL methods have been shown to reduce the need for environment interaction by training agents using offline collected episodes. However, the action information in offline episodes can be difficult or even impossible to collect in some practical cases. This paper investigates the problem of using action-free offline datasets to improve online reinforcement learning. We introduce Action-Free Guide (AF-Guide), a method to extract task-relevant knowledge from separate action-free offline datasets. AF-Guide employs an Action-Free Decision Transformer (AFDT) that learns from such datasets to plan the next states, given desired future returns. In turn, AFDT guides an online-learning agent trained by "Guided Soft Actor-Critic" (Guided SAC). Experiments show that AF-Guide can improve RL sample efficiency and performance. Our code is in the supplementary and will be made publicly available.

## 1 Introduction

Training a reinforcement learning (RL) agent from scratch can be a challenging task that requires time-consuming exploration of its environment. Offline RL methods pretrain RL agents via offline collected episodes to reduce the number of interactions with the environment in the subsequent online fine-tuning stage. Traditional offline RL methods, however, require offline data with action labels (e.g., motor torques), which can be difficult or even impossible to collect in practice. Nevertheless, even "action-free" data that lacks action labels may hold valuable algorithmic information about consequences of agent movements (e.g., environmental transitions) (Schmidhuber, 2015; 2018), which should be extracted to facilitate the RL agent's task.

In this paper, we utilize action-free offline reinforcement learning datasets to guide online RL and name this setting RL with Action-Free Offline Pretraining (AFP-RL). We propose Action-Free Guide (AF-Guide), a method that improves online training by learning to plan good target states from action-free offline datasets. AF-Guide comprises two main components: an Action-Free Decision Transformer (AFDT), and a Guided Soft Actor-Critic (Guided SAC). AFDT, a variant of the Upside Down RL (Schmidhuber, 2019) model Decision Transformer (Chen et al., 2021), is trained on an offline dataset without actions to plan the next states based on the past states and the desired future returns. Guided SAC, a variation of SAC (Haarnoja et al., 2018a;b), follows the planning of AFDT by maintaining an additional Q function that fits a guiding reward built from the negative discrepancy between the planned state and the achieved state with zero discount factor. An overview of our method is summarized in Fig.1. Our experimental results demonstrate that AF-Guide can significantly improve sample efficiency during online training by utilizing action-free offline datasets.

Our contribution can be summarized as follows:

- We propose RL with Action-Free Offline Pretraining (AFP-RL), a novel setting to study how to guide online RL with offline datasets that do not contain explicit action labels.

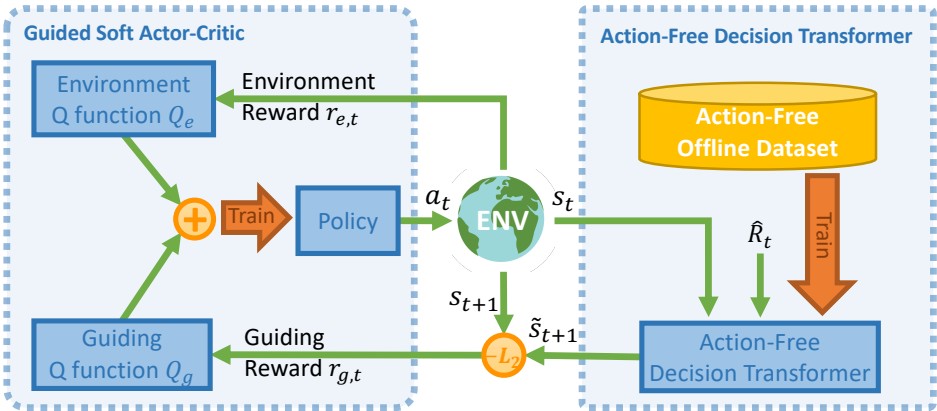

Figure 1: An overview of AF-Guide. Action-Free Decision Transformer (AFDT) is trained on the action-free offline dataset to plan the next state $\tilde{s}_{t+1}$ given previous states and the desired return-to-go $\hat{R}_t$. The guiding reward $r_g$ is formed based on the negative L2 distance between the planned state $\tilde{s}_{t+1}$ and the real state $s_{t+1}$. In addition to SAC's original Q function denoted as $Q_e$ that fits the environment reward $r_e$, Guided SAC has an additional Q function $Q_g$ to fit the guiding reward $r_g$ with zero discount factor to discard the future return. The policy is trained by the weighted sum over the two Q functions.

- We present Action-Free Guide (AF-Guide), a method that pretrains a model which can extract knowledge from the action-free offline dataset and conduct state-space planning to guide online policy learning.

- Experimental results show that AF-Guide can benefit from the action-free offline dataset to improve sample efficiency and performance during online training.

## 2    Related Work

**Offline Reinforcement Learning**    Offline reinforcement learning methods learn policies using pre-collected episodes from unknown behavior policies. Many offline RL methods, such as CQL (Kumar et al., 2020), IQL (Kostrikov et al., 2021), AWAC (Nair et al., 2020), BCQ (Fujimoto et al., 2019), and COMBO (Yu et al., 2021), have been developed from off-policy algorithms, with additional constraints to avoid out-of-distribution actions that are not covered by the dataset. Recently, Decision Transformer (Chen et al., 2021) and Trajectory Transformer (Janner et al., 2021) convert the offline RL problem as a context-conditioned sequential modeling problem and generate good actions by either conditioning on desired future return following Upside Down Reinforcement Learning framework (Schmidhuber, 2019) or searching for a good rollout with a high future return. In our AFP-RL setting, datasets do not contain actions. In this case, learning an offline policy directly is infeasible. Our method AFDT-Guide instead leverages action-free data to plan good target states and guide online training for improved performance.

**Imitation Learning from Observation**    The target of imitation learning from observation is to learn a policy through state-only action-free demonstrations from experts. imitation Learning from observation methods can be broadly classified into different categories. Methods like GSP (Pathak et al., 2018) and BCO (Torabi et al., 2018a) train an inverse dynamic model to infer the expert actions given state transitions. Reward-based methods like DeepMimic (Peng et al., 2018) and Context-Aware Translation (Liu et al., 2018) create surrogate reward functions to guide online training. Other methods like GAIfO (Torabi et al., 2018b), IDDM (Yang et al., 2019), MobILE (Kidambi et al., 2021) employ adversarial learning. The difference between imitation learning from observation and our setting AFP-RL is similar to the difference between imitation learning and offline RL. In imitation learning from observation, the dataset is collected by an expert policy, and agents are trained to directly imitate the collected episodes. In contrast, episodes in AFP-RL are

collected by behavior policies that may be suboptimal. As a result, directly imitating these episodes would lead to suboptimal performance.

**Motion Forecasting**   Motion forecasting is the task of predicting the future motion of agents given past and context information. It helps autonomous systems like autonomous driving and robotics to foresee and avoid potential risks like collisions in advance. Recent methods for motion forecasting have explored various architectural designs. For example, Social-LSTM (Alahi et al., 2016) and Trajectron++ (Salzmann et al., 2020) are based on RNN. Social-GAN (Gupta et al., 2018) and HalentNet (Zhu et al., 2021) benefit from generative adversarial training. Social-STGCNN (Mohamed et al., 2020) and Social-Implicit (Mohamed et al., 2022) predict the future via spatial-temporal convolution. AgentFormer (Yuan et al., 2021), mmTransformer (Liu et al., 2021), and ST-Transformer (Aksan et al., 2021) are models based on Transformer architecture (Vaswani et al., 2017) designed for pedestrian or vehicle trajectory prediction. Our state-planner AFDT is a Transformer model. Instead of simply predicting the future states conditioned on the past, AFDT plans the future states by additionally conditioned on the desired future return.

## 3   Background

**Soft Actor-Critic (SAC)**   SAC is an actor-critic RL approach based on the maximum entropy framework (Haarnoja et al., 2018a;b), which involves optimizing a Q network $Q_e$ [1] and the policy network $\pi$. The Q function $Q_e$ is learned with the following objective

$$\min_{Q_e} E_{\mathcal{D}_{\text{online}}} \| Q_e(s_t, a_t) - Q_{e,t}^{\text{target}} \|_2^2 \tag{1}$$

where $\mathcal{D}_{\text{online}} \triangleq \{(s_t, a_t, r_{e,t}, s_{t+1})\}$ with state $s_t$, action $a_t$, environment reward $r_{e,t}$, and next state $s_{t+1}$, is the online replay buffer. $Q_{e,t}^{\text{target}}$ is the target Q value computed as follows

$$Q_{e,t}^{\text{target}} = r_{e,t} + \gamma \mathbb{E}_\pi \left[ Q_e(s_{t+1}, a_{t+1}) - \alpha \log \pi(a_{t+1}|s_{t+1}) \right] \tag{2}$$

Here, $\gamma$ is the discount factor and $\alpha$ is the temperature parameter to weight the entropy. The policy network is learned by the following objective

$$\min_\pi \mathbb{E}_{s_t \sim \mathcal{D}_{\text{online}}, a_t \sim \pi} \left[ \alpha \log(\pi(a_t|s_t)) - Q_e(s_t, a_t) \right] \tag{3}$$

**Upside Down Reinforcement Learning and Decision Transformer**   Traditional Reinforcement Learning methods are trained to predict future rewards (e.g., a Q function) first and convert the prediction into rewarding actions. In contrast, Upside Down Reinforcement Learning (UDRL) (Schmidhuber, 2019) framework takes desired future rewards as inputs to generate actions. As an instance of UDRL in the offline RL setting, Decision Transformer (DT) (Chen et al., 2021) is trained in the offline dataset to regress the current action $a_t$ conditioned on the past $K$ states $s_{t-k:t}$, actions $a_{t-k:t-1}$, and the future returns (named Return-To-Go, RTG) $\hat{R}_{t-k:t}$, with $\hat{R}_t = \sum_{t'=t}^T r_{t'}$. The architecture of DT is based on the language model GPT (Radford et al., 2018). When evaluated in an environment, the model is provided with an initial state $s_0$ and a desired initial RTG $\hat{R}_0$ to generate the first action. After executing the action $a_t$ in the environment and observing the reward $r_t$ and the next state $s_{t+1}$, the RTG is updated by $\hat{R}_{t+1} = \hat{R}_t - r_t$. The executed action $a_t$, the current return-to-go $\hat{R}_{t+1}$, and the current state $s_{t+1}$ are then fed back into DT to infer the next action. Given a high initial RTG $\hat{R}_0$, DT is able to generate good actions that lead to high future returns. Due to the dependence of standard DT on action labels, it can not be directly applied for Action Free Pretraining.

## 4   Action-Free Guide

**Action-Free Offline Pretraining**   In the setting of Reinforcement Learning with Action-Free Offline Pretraining (AFP-RL), an action-free offline dataset, $\mathcal{D} = \{\tau_1, \tau_2, ..., \tau_N\}$, is provided to boost the online

---

[1]We use the subscript $e$ to denote notations related to the environment reward, and will use $g$ to differentiate the notations related to the guiding reward (see 4.2).

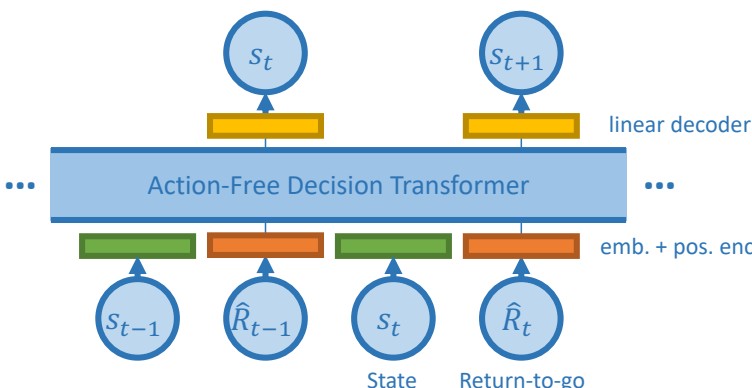

Figure 2: Action-Free Decision Transformer. The next state is planned given previous states and a desired return-to-go.

training in the environment. The trajectories in the dataset have been pre-collected in the environment by behavior policies that are unknown to the agent. Each trajectory, $\tau$, contains states and rewards in the format $\tau = (s_0, r_0, s_1, r_1, ..., s_T, r_T)$, with $T$ time steps. Unlike traditional offline RL, where the policy is learned directly from the offline dataset, it is infeasible to learn a policy from an action-free offline dataset as it lacks the necessary action information. However, such a dataset still contains valuable information about the agent's movements and the environment's dynamics. Our proposed setting, AFP-RL, aims to leverage this information to improve online training.

**Methodology Overview**  Our method, Action-Free Guide (AF-Guide), utilizes knowledge from action-free offline datasets by training an Action-Free Decision Transformer (AFDT) on these datasets to plan the next states that lead to high future returns. Then, the online agents, trained by Guided Soft Actor-Critic (Guided SAC), follow the planning with an additional Q function optimized for a new guiding reward. The overall methodology is illustrated in Fig.1.

## 4.1 Action-Free Decision Transformer

Action-Free Guide(AF-Guide) can be considered as a variant of the UDRL model Decision Transformer (DT) (Chen et al., 2021) that we designed to operate on action-free offline datasets. Unlike DT, which predicts actions based on past RTGs, states, and actions, AFDT plans the next state based on previous states and RTGs only. The overall architecture of AFDT is illustrated in Fig.2. AFDT takes $K$ steps of input, consisting of $2K$ tokens, where each step contains a state and an RTG. Similar to DT, states and RTGs are first mapped to token embedding via separate single-layer state and return-to-go encoders $Embed_s$ and $Embed_R$. The positional embedding mapped from time steps $t$ by a single-layer temporal encoder $Embed_t$ is then added to the token embedding to include temporal information, followed by layer normalization. These token embeddings are then processed by a GPT model (Radford et al., 2018). The next states are generated from the processed RTG tokens through a single-layer decoder $Pred_s$. Note that we don't predict the next state $s_{t+1}$ directly, but rather predict the state change $\Delta s_{t+1} = s_{t+1} - s_t$ first and add it back to $s_t$ to obtain $s_{t+1}$. This is a common practice in motion forecasting (e.g., Mohamed et al. (2020); Salzmann et al. (2020)) to improve the prediction accuracy and has been observed to improve the performance of our model in experiments. The algorithm of AFDT is listed in Algo.1.

**Training**  At each training step, a batch of trajectories truncated to length $K$ is randomly sampled from the dataset. Each trajectory contains states and precomputed RTGs, represented as $\tau = (s_{t-K+1}, \hat{R}_{t-K+1}, ..., s_t, \hat{R}_t)$. The model is trained autoregressively with L1 loss to predict the next state from the processed RTG token at each time step, using a causal mask to mask out future information.

---

**Algorithm 1** Action-Free Guide

---

**Input:** states $s$, returns-to-go $\hat{R}$, time steps $t$, temporal encoder $Embed_t(\cdot)$, state encoder $Embed_s(\cdot)$, return-to-go encoder $Embed_R(\cdot)$, state decoder $Pred_s(\cdot)$

  *# get positional embedding for each time step*
  $f_t = Embed_t(t)$
  *# compute the state and return-to-go embeddings*
  $f_s, f_{\hat{R}} = Embed_s(s) + f_t,\ Embed_R(\hat{R}) + f_t$
  *# send to transformer in the order $(s_0, \hat{R}_0, s_1, \hat{R}_1, ...)$*

  $f_{output} = \text{Transformer}(\text{stack}(f_s, f_{\hat{R}}))$
  *# predict the state change*
  $\Delta s = Pred_s(\text{unstack}(f_{output}.\text{states}))$
**Output:** $\Delta s + \text{s}$

---

**Algorithm 2** Compute Guiding Reward

---

**Input:** states $s_{1:t}$, return-to-go $\hat{R}_{1:t}$, policy $\pi$, state standard deviation $\sigma_{\mathcal{D}}$, environment $env$, AFDT with context length $K$
**repeat**
  *# get AFDT's prediction of the next state*
  $\widetilde{s}_{t+1} = \text{AFDT}(s_{t-K+1:t}, \hat{R}_{t-K+1,t})$
  *# apply the policy in the environment for one step*

  $a_t = \pi(s_t)$
  $s_{t+1}, r_e = env.\text{step}(a_t)$
  *# compute current guiding reward using Eq.4*
  $r_g = -\|\frac{1}{\sigma_{\mathcal{D}}} \odot (\widetilde{s}_{t+1} - s_{t+1})\|_2$
  *# update return-to-go (same as DT) and time step*

  $\hat{R}_{t+1} = \hat{R}_t - r_e$
  $t = t + 1$
**until** Episode is finished

---

## 4.2 Guided Soft Actor-Critic

Now we illustrate how to use the AFDT model to benefit the learning of Soft Actor-Critic (SAC) (Haarnoja et al., 2018a;b). As AFDT can conduct planning in the state space and infer the subsequent states that lead to a high future return, our idea is to guide the agent to follow AFDT's planning. Our method, named *Guided SAC*, contains the following three main procedures.

**Guiding Reward** We first design a *guiding reward* $r_{g,t}$, which is the negative discrepancy between the planned state $\widetilde{s}_{t+1}$ inferred by AFDT and the actual state $s_{t+1} \sim \text{P}(\cdot|s_t, a_t)$ achieved by the agent:

$$r_{g,t} = -\|\frac{1}{\sigma_{\mathcal{D}}} \odot (\widetilde{s}_{t+1} - s_{t+1})\|_2 \tag{4}$$

where $\sigma_{\mathcal{D}}$ is the standard deviation of the states over the entire offline dataset $\mathcal{D}$ and is used to normalize the different-scale state values on different dimensions. By maximizing the guiding reward $r_{g,t}$, the policy is encouraged to reach states that are close to AFDT's planning. The process to compute the guiding reward with the AFDT model is summarized in Algo.2.

**Guiding Q Function** We then use the guiding reward $r_{g,t}$ to learn the Q function. A common practice to include a new reward is simply adding the new reward to the original environment reward $r_{e,t}$ with a coefficient $\beta$, like $r_t = r_{e,t} + \beta r_{g,t}$, and use a single Q network $Q$ to approximate the long-term future return (Schmidhuber, 1990; 1991; Houthooft et al., 2016; Pathak et al., 2017; Tao et al., 2020). However, this is not the case for the guiding reward, where *the current action should only be responsible for the next immediate result rather than all the future results*. Assume a robot gets stuck at step $t + 1$ due to a bad action $a_t$ at step $t$. A good AFDT will give the robot a low guidance reward at step $t$ and predict a static future, resulting in high future guidance rewards for getting the robot stuck in the same state. More generally, as AFDT replans the target states at every timestep, an agent missing the planned state $\tilde{s}_t$ due to a bad action $a_{t-1}$ can still reach the replanned state $\tilde{s}_{t+1}$ at the next step and receive a high guiding reward $r_{g,t}$, which is not desirable. Hence, the action $a_t$ should not be rewarded by $r_{g,t+1}$ as it didn't reach the original plan $\tilde{s}_t$. Therefore, to prevent the guiding reward from misleading the agent, it is more reasonable to discard the future return for the Q value calculation of the current action.

Due to the reason above, we set up an additional independent Guiding Q function $Q_g$ which is optimized in the same way as the original Q function $Q_e$ (see Eq.1), but the target Q value only involve the immediate reward $r_{g,t}$ without future information, which is computed as follows:

$$Q_{g,t}^{\text{target}} = r_{g,t} \tag{5}$$

Compared to Eq.2, here the Q target of the current action is removed from the future information by setting the discount factor $\gamma$ to zero. Our ablation study in the experiment section demonstrates that the Guiding Q function is crucial to effective guidance.

**Combined Q function**   We finally replace the Q function $Q_e$ in Eq.3 with the following combined Q function to guide the policy learning:

$$Q(s_t, a_t) = Q_e(s_t, a_t) + \beta Q_g(s_t, a_t) \tag{6}$$

where $\beta$ is the coefficient. Note that when $\beta = 0$, Guided SAC degenerates to a standard SAC trained using environment rewards $r_e$ and the corresponding Q function $Q_e$ only.

## 5   Experiments

In this section, we demonstrate the effectiveness of our approach AF-Guide for utilizing action-free offline reinforcement learning datasets in online reinforcement learning through experimental evaluation. Furthermore, we provide evidence for the validity of our design choices for the two components of AF-Guide, Action-Free Decision Transformer, and Guided SAC, through three ablation studies.

**Action-Free D4RL Benchmark**   To evaluate methods on AFP-RL, we adapt the widely-used offline RL benchmark D4RL (Fu et al., 2020) to the action-free reinforcement learning setting and denote it as Action-Free D4RL. The original D4RL benchmark provides offline datasets collected using various strategies across different environments. These episodes in the original D4RL datasets include state, action, and reward sequences. To create our action-free offline RL datasets, we remove the action labels from the original datasets. We evaluate six environments, including three locomotion tasks (Hopper, Halfcheetah, Walker2d), two ball maze environments (Maze2d-Medium, Maze2d-Large), and one robot ant maze environment (Antmaze-Umaze). For each locomotion task, we test our method on three different datasets: Medium, Medium-Replay, and Medium-Expert. For the environment Antmaze-Umaze, we test on two datasets: Antmaze-Umaze and Antmaze-Umaze-Diverse. There is only one dataset for each ball maze environment, where the ball navigates to random goal locations. Details of the datasets can be found in the supplementary.

**Implementation Details**   The training of AF-Guide contains two stages: an offline stage training AFDT using the offline dataset and an online stage training Guided SAC in the environment. We follow the default hyperparameters used in DT paper (Chen et al., 2021) for AFDT. The context length $K$ is set to 20. The batch size for AFDT training is 64 and the learning rate is 1e-4 with AdamW optimizer. In the online training stage, we set RTG $\hat{R}$ to 6000, 3600, and 5000 for Halfcheetah, Hopper, and Walker2d, respectively, the same as the values used in the original DT paper. The robot ant maze environment and the ball maze environments are not used in the original DT paper. We set $\hat{R}$ to 1 and 5000, separately. For the hyperparameters of Guided SAC, we follow the default setting of SAC in the widely used Stable Baseline 3 (Raffin et al., 2021) implementation. The batch size is 256 and the learning rate is 3e-4 with Adam optimizer. The discount factor for the environment reward is 0.99. The coefficient of the Guided Q function $\beta$ in Eq.6 is set to 3. More details can be found in the supplementary.

### 5.1   Main Experiments

Experimental results are presented in Fig.3. We run each experiment four times and report the average and the standard deviation band. Our method AF-Guide , using knowledge learned from the action-free offline dataset, outperforms SAC in all the evaluated environments. In the tasks of Halfcheetah and Walker2d, AF-Guide shows a significant advantage in learning speed compared to SAC across all three datasets. In Halfcheetah, AF-Guide demonstrates a significant improvement of 50% at 500k steps with a performance of 6000 compared to 4000 achieved by SAC alone. Similarly, in Walker2d, AF-Guide improves the performance by 50% at 1M steps, from 2000 to 3000. Additionally, we observe that different offline datasets do not result in significant performance differences. In the tasks of Hopper, Maze2d-Medium, and Maze2d-Large, while both AF-Guide and SAC reach similar performance at 500k steps, AF-Guide converges faster. In the

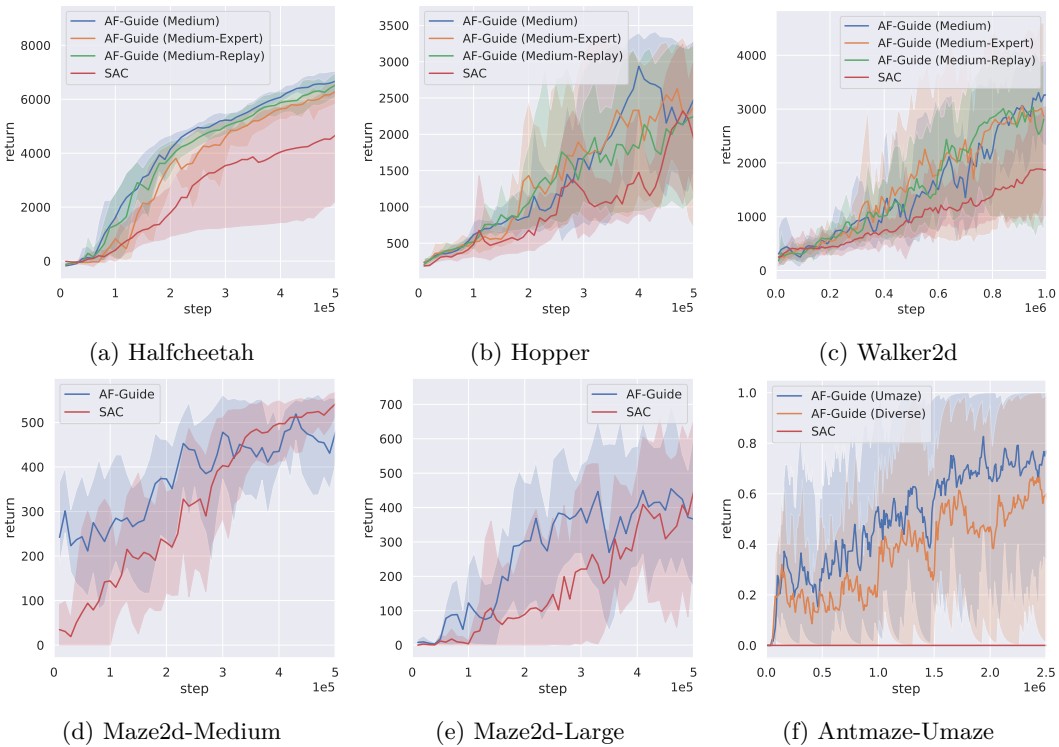

Figure 3: Experimental results of our methods. Utilizing the knowledge learned from the action-free offline dataset, AF-Guide outperforms SAC in all evaluated locomotion and ball maze environments in terms of learning speed. Furthermore, while SAC struggles to complete the task of Antmaze-Umaze due to the challenging exploration, AF-Guide successfully solves it, owing to the guidance signals provided by AFDT.

task of Antmaze-Umaze, SAC is unable to complete it in 1M steps, whereas AF-Guide has an 80% success rate when pretrained in the dataset Antmaze-Umaze and a 60% success rate in Antmaze-Umaze-Diverse. This is likely due to the exploration challenge faced by SAC. The robot ant in Antmaze-Umaze has large state/action spaces with 8 joints and only receives sparse rewards when reaching the target location. Thus, the agent trained by SAC rarely receives any rewards during exploration. In contrast, our guiding reward provides dense learning signals that guide the agent's motion toward the target. Therefore, agents trained by AF-Guide can successfully solve the maze here.

## 5.2 Ablation Study

**Do we really need Guided SAC?** Here, we investigate whether our Guided SAC with an additional Q function is necessary to process the guiding reward $r_g$, or if it can be simply added to the environment reward and processed by SAC, referred to as 'AF-Guide [SAC]'. This study is conducted in the locomotion environments of Halfcheetah and Walker2d using the Medium dataset and the maze environment of Maze2d-Medium. The results in Fig.4 reveal that AF-Guide [SAC] performs similarly to SAC in Maze2d-Medium and does not work in Halfcheetah and Walker2d, indicating that the guiding reward $r_g$ does not help or even hinders the training of SAC. In contrast, AF-Guide with Guided SAC benefits from the guiding reward $r_g$ by ignoring guiding rewards in future steps and setting the corresponding discount factor to zero. This is in line with our explanation in the guided soft actor-critic section that high future guiding rewards are unrelated to the current action and should be ignored in the Q function and verifies the effectiveness of our Guided SAC design.

**Does AFDT plan better states than that from behavior policy?** Here, we first trained an AFDT in an 'imitation' style, denoted as AF-Imitation, by regressing future states without any RTG information

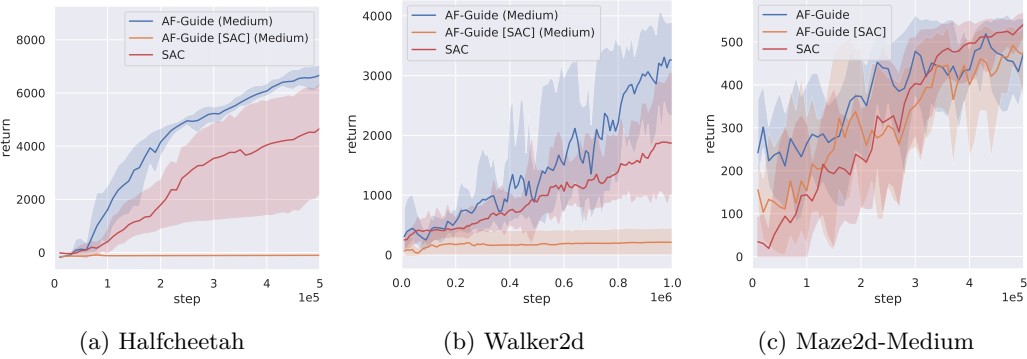

(a) Halfcheetah      (b) Walker2d      (c) Maze2d-Medium

Figure 4: Ablation study on the guiding reward $r_g$. 'AF-Guide [SAC]' denotes the variant adding guiding reward to the environment reward and training with SAC. The results show that AF-Guide [SAC] performs similarly to SAC in Maze2d-Medium, but does not work in Halfcheetah and Walker2d, which indicates that simply adding the guiding reward is detrimental to the policy training and verifies the effectiveness of our Guided SAC design.

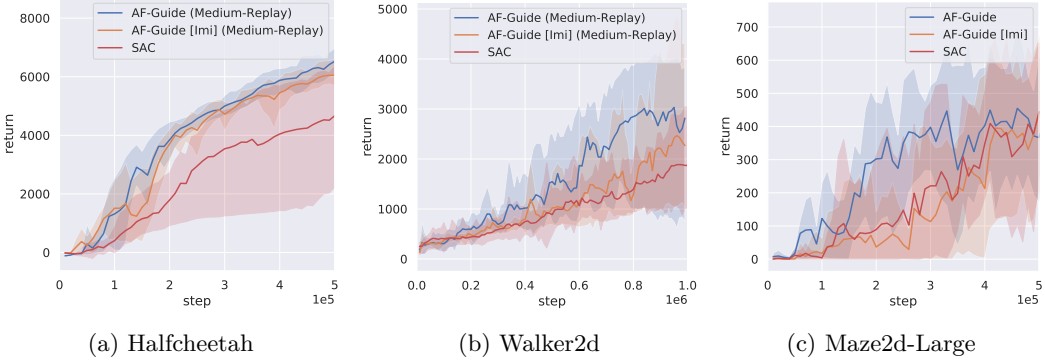

(a) Halfcheetah      (b) Walker2d      (c) Maze2d-Large

Figure 5: Ablation study on the effectiveness of Action-Free Decision Transformer (AFDT). We train a variant of AFDT by regressing the behavior policy trajectories and use this variant to guide the online training, referred to as AF-Guide [Imi]. Compared to AF-Guide, AF-Guide [Imi] performs worse in Walker2d and Maze2d-Large. This suggests that AFDT can infer the next states better than those collected by the behavior policy.

in the offline dataset. Then, we learn a policy with the guidance of AF-Imitation, named 'AF-Guide [Imi]', and evaluate it in Halfcheetah and Walker2d with the Medium-Replay dataset and also in Maze2d-Large. Fig.5 demonstrates that AF-Guide [Imi] underperforms the original version in Walker2d and Maze2d-Large, indicating that AFDT plans better next states than the behavior policy when conditioned on a proper RTG. Additionally, AF-Guide [Imi] outperforms SAC in Halfcheetah and Walker2d, showing that AF-Imitation still benefits policy training in some cases despite the suboptimal planning.

**Can Action-Free Trajectory Transformer replaces Action-Free Decision Transformer?** Guided SAC is built on the predictions of AFDT, our action-free variant of the Decision Transformer (DT). In theory, AFDT can be replaced by any other sequential-modeling-based offline RL method after removing the action information. Here, we replaced AFDT with an action-free variant of Trajectory Transformer (TT) (Janner et al., 2021) and evaluated it on locomotion tasks with the Medium dataset. We denote this variant as AF-Guide [TT]. Details of AF-Guide [TT] is in the supplementary materials. Compared to DT which plans in a UDRL style, TT rollouts the future via beam search and selects the highest return one. Additionally, TT discretizes the state and action spaces to improve prediction accuracy. Results are shown in Fig.6. AF-Guide [TT] performs worse than AF-Guide in Halfcheetah but shows a clear advantage over

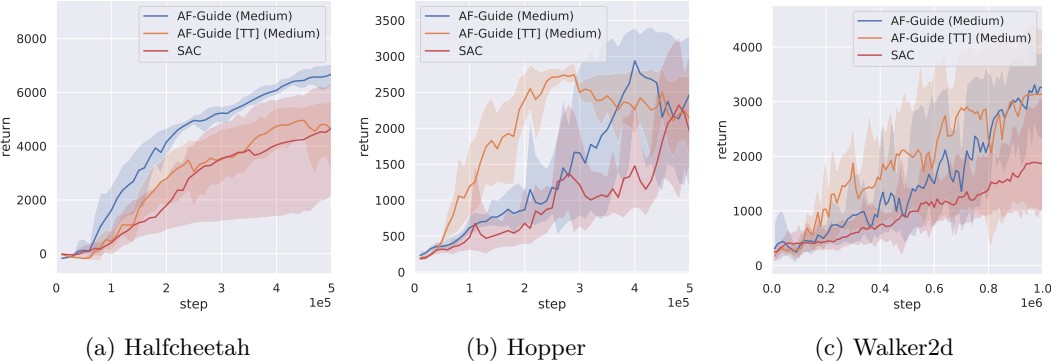

(a) Halfcheetah   (b) Hopper   (c) Walker2d

Figure 6: Ablation study on using Action-Free Trajectory Transformer to guide the training (AF-Guide [TT]). The results showed that AF-Guide [TT] had better performance in the Hopper and Walker2d tasks, but performed worse in the Halfcheetah task. These results suggest that our pipeline is compatible with different sequential-modeling-based offline RL methods, but the choice of method may impact performance depending on the specific task.

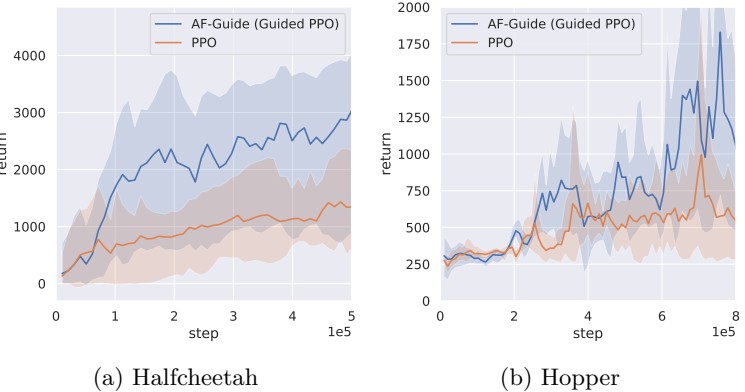

(a) Halfcheetah   (b) Hopper

Table 1: Training time of AF-Guide [TT] and AF-Guide for 500k steps in locomotion tasks on one A100 GPU. AF-Guide [TT] increases the training time dramatically due to the huge planning cost.

| Env. | AF-Guide [TT] | AF-Guide |
|---|---|---|
| Halfcheetah | ~14 hours | ~1 hour |
| Hopper | ~10 hours | ~1 hour |
| Walker2d | ~20 hours | ~1 hour |

Figure 7: Combining AF-Guide with PPO speeds up the training in the test environments Halfcheetah and Hopper.

AF-Guide in Hopper and Walker2d. This advantage of AF-Guide [TT] in Hopper and Walker2d may be due to the better prediction quality from the discretization.

However, AF-Guide [TT] has a much longer training time due to the huge planning cost for the discretization and the beam search. A brief training time comparison between AF-Guide and AF-Guide [TT] is shown in Tab.1. AF-Guide [TT] is at least 10 times slower than AF-Guide in our experiments. Therefore, we use DT in our final design. Experiments also show that our pipeline is compatible with different sequential-modeling-based offline RL methods.

**Can AF-Guide be applied to other RL methods besides SAC?** Theoretically, AF-Guide can be combined with other online RL algorithms that also have Q/value functions. In this ablation study, we combine AF-Guide with PPO (Schulman et al., 2017) and name it Guided PPO. Similar to Guided SAC, we introduce an independent guiding value function $V_g$ where the target value only involves the immediate guiding reward $r_{g,t}$ with zero discount factor. Experimental results in Fig.7 show that Guided PPO outperforms PPO in the test environments Halfcheetah and Hopper, demonstrating the effectiveness of the guiding signals from AF-Guide in PPO.

### 5.3 Limitations

As an attempt to utilize action-free offline datasets for improved online learning, AF-Guide has some limitations in its current form. Firstly, AF-Guide 's current planning ability is limited by Decision Transformer. As AF-Guide is agnostic to the planning model shown in our ablation study with Trajectory Transformer, we believe AF-Guide can benefit from a more powerful planning model in the future. Secondly, the current guiding reward is based on L2 distance, which may not be optimal in some state spaces where L2 distance is not the best similarity metric, such as images. We believe that combining AF-Guide with more semantically meaningful similarity metrics can extend its applications for vision, language, and other multimodal problems in the future.

## 6 Conclusion

Our Action-Free Offline Pretraining (AFP-RL) improves online reinforcement learning (RL) by exploiting "action-free" offline datasets that do not contain *explicit* labels of the actions executed by observed agents (although they may contain *implicit* information about the consequences of such actions). Our Action-Free Guide (AF-Guide) learns from such datasets to plan goal-conditional target states, thus guiding online RL agents. Our experimental results demonstrate that AF-Guide yields better sample efficiency than the Soft Actor Critic in various locomotion and maze environments, highlighting the benefits of incorporating action-free offline datasets. We hope our work will encourage further research on action-free offline pre-training.

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

# A    Appendix

## A.1    Datasets

For locomotion environments, the Medium dataset is collected using a policy trained to approximately 1/3 the performance of an expert. Medium-Replay uses the training replay buffer of the 'Medium' policy. The Medium-Expert dataset contains 50% of data from Medium and the remaining data is collected by an expert policy. For the ant robot maze environment, in the Antmaze-Umaze dataset, the robot ant always goes from a fixed start position to a fixed target location, while in Antmaze-Umaze-Diverse, the robot ant goes to random target locations.

## A.2    Hyperparameters

For the architecture of AFDT, we follow the default hyperparameters of DT. In detail, we use three transformer blocks for most of the environments and one for ball maze environments. Each block has one attention head. The embedding dimension is set to 128. Dropout rate is set to 0.1. We train AFDT for 50000 gradient steps and selected the best checkpoint from 3000 steps, 5000 steps, 10000 steps, 15000 steps, 30000 steps and 50000 steps. The implementation of AFDT is based on the repository 'minimal decision transformer'(Barhate, 2022). For the architecture of Guided SAC, the environment Q function, the guided Q function, and the policy net are all three-layer MLPs with ReLU activation function and 256 hidden dimensions. The implementation of Guided SAC is based on the repository 'Stable Baseline 3'(Raffin et al., 2021).

## A.3    Action-Free Trajectory Transformer

Our implementation of AFTT is based on the repository 'faster-trajectory-transformer' (Nikulin, 2022), which has an improved inference speed compared to the original implementation. We follow all the default hyperparameters and model architectures in the repository, but remove the action-related component in TT to build AFTT. We use a uniform strategy to discretize the state space.

