# OpenReview forum: "Guiding Online Reinforcement Learning with Action-Free Offline Pretraining"
_TMLR — Withdrawn by Authors_

### Review · Reviewer_CRAn · 2023-12-12

**Summary Of Contributions:**

The authors want to find a way to improve an RL algorithm given a dataset of off-policy interactions in the environment, without action data (i.e. sequences of $(s, r, s, r, ...)$).

The proposed method to do so is to first fit a Decision Transformer, predicting next state $s_{t+1}$ given $s_t$ and return to go $R_t$. Once learned, this is used to define a "guiding reward". The Action-Free Decision Transformer is conditioned on high return to go, then asked to predict next states, which can be interpreted as outputting goal next-states that are more likely in high-return trajectories. The guiding reward is based on the distance between true $s_t$ and predicted $\hat{s}_t$, and is maximized when $||s_t - \hat{s}_t||^2$ is minimized.

Although this is described as a guiding reward, the authors argue that we do not want to maximize the cumulative guiding reward over the trajectory. An example given is that if the agent takes an action that makes future states very predictable (i.e. robot is stuck), then a small guiding reward $r_{g,t}$ at time = $t$ will lead to large $r_{g,t+k}$ for all future timesteps.

So, when used in SAC, the actor is trained to maximize (entropy) + (Q-value) + beta * (guiding reward *for current timestep t*)

**Audience:**

Yes

**Claims And Evidence:**

No

**Requested Changes:**

Please add more baselines + cite more related work. I do not think the paper is sufficiently comparing itself to other options - it is only comparing variants of AF-Guide to ablations of AF-Guide and the absence of AF-Guide.

**Strengths And Weaknesses:**

The problem setting is interesting, but I'd say the paper is lacking in other baselines for how to use action-free off-policy data.

Why not use the action-free data as a pretraining step for future SAC? Something like, fit a model to learn how to predict states and then init from those weights for another method. See Seo et al, "Reinforcement Learning with Action-Free Pre-Training", ICML 2022, which does something similar to this with more tricks. (This was literally the first search result for me on "action-free reinforcement learning", making it a bit weird it wasn't discussed at all.)

Alternatively, the paper could be sharing weights between the AFDT and policy $\pi$, using the AFDT objective just as an auxiliary task to improve features learned for policy $\pi$, and then not directly using the AFDT prediction for any part of the SAC training. This would cast it more like the Jaderberg et al "Reinforcement Learning with Unsupervised Auxilairy Tasks" 2016 paper.

Or, considering the connections to offline RL, the CQL paper adds a conservatism term for actions $a$ that do not appear in the dataset, by minimizing $Q(s,a)$ for said actions $a$. You could imagine a similar conservatism term here - except, instead of minimizing $Q(s,a)$ if $a \not\in dataset$, it would instead be "minimize $Q(s,a)$ if observed $s_{t+1}$ from online learning is not in the dataset".

It's just not clear to me that the guidance should specifically be used as an additive reward, rather than other schemes of incorporating the action-free data.

---

### Review · Reviewer_YqJd · 2024-02-05

**Summary Of Contributions:**

A problem setting this paper considers is a setup where action-free datasets are available without action labels (but with reward labels). In this setup, the paper presents a new method that trains a Transformer model that predicts the next state, and designs a reward that encourages the agent to visit the desired next state. This desired next state is predicted by the Transformer model conditioned on high return-to-go, thus letting the agent visit the high-return states. Experiments are conducted in D4RL benchmark.

**Audience:**

No

**Claims And Evidence:**

No

**Requested Changes:**

- Can the authors clarify the limitations from the assumptions made in the paper, and include a discussion on what the paper exactly aims to achieve? I would not say that the paper should also consider the setup without the rewards as a TMLR submission, but the paper should make it clear what the aim of this paper is.
- Can this paper's method be extended to visual domains? How could this be extended to images when correctly predicting the future image still remains a challenge? At least showing some results on simple visual domains (where it is known that dynamics models can generate high-fidelity future frames, such as GridWorld or simple DMC environments) could be helpful to show the promise of this approach for more complex domains.
- Can the author discuss what's the implication of having a myopic intrinsic reward on the convergence of Q functions? Would it converge? How does it affect asymptotic performance?
-  Crucial baselines are several works introduced in 'Imitation Learning from Observation' section from the related work. Can these baselines be included?

**Strengths And Weaknesses:**

Strengths
- The idea of better utilizing action-free datasets is very important and timely for improving the sample-efficiency of online RL algorithms.
- The proposed idea makes sense and the writing is easy-to-follow to help understanding the core idea
- Experiements on various design choices of the proposed method.

Weaknesses
- The paper is depending on a heavy assumption that rewards are available from the action-free datasets.
- The paper is missing thorough investigation/discussion on how the learned Q-value converges with one-step intrinsic reward
- The paper does not investigate the asymptotic performance of RL agents trained with the intrinsic reward; what would happen when the agents get better than the behavior policies in action-free datasets?
- Comparison against state-only imitation learning approaches are missing from the paper

---

### Review · Reviewer_Rv96 · 2024-03-14

**Summary Of Contributions:**

The authors propose a method to improve an online RL algorithm using a dataset of demonstrations with no actions.

They first train a Decision Transformer to predict next states in the offline dataset. Then, they use the AFDT to compute a reward that will help the downstream online agent. This reward is simple to understand intuitively, as it is a difference between the predicted state and the actually achieved state: it will encourage the agent to follow demonstrations from the dataset.

The method is then tested on Mujoco environments and demonstrates improvement against SAC.

**Audience:**

Yes

**Broader Impact Concerns:**

No impact statement is needed.

**Claims And Evidence:**

No

**Requested Changes:**

I think fixing the weaknesses 1, 2 and 3 would greatly improve the paper.

About weakness 1: while the comparsion may be done fairaly withing the same codebase, and I understand that reproducing SAC results is not an easy task, I would strongly suggest using an implementation of SAC that at least gets closer to its reported "official" performance. This is currently the only reason why I answered "no" at the claims and evidence review question.

Weaknesses 2 and 3 are mostly about writing.

Weaknesses 4 and 5 are, I think, interesting research questions, but I also do not think they are a priority to fix.

**Strengths And Weaknesses:**

# Strengths

The ideas in the paper are easy to grasp and well laid out. The principle is simple but very general: it could be adapted to different setups, with different transformers design, reward design and online RL algorithms.

The setup is also interesting: to my knowledge, using action-free demonstrations is not a widely explored area of research.


# Weaknesses

The weaknesses are listed in order of importance; from important to details.


### 1/ Quality of baselines

The quality of SAC does not seem on par with sota implementations of the method. For example, on HalfCheetah, [4] reports a score of 10k at 500k steps (compared to less than 6k reported in this work). Additionally, at this step range, SAC is clearly far from convergence, and it would be worthwhile to continue training at least up to 1 million training steps. Right now, it is difficult to trust the results of the paper with such a discrepancy in empirical performance.

[4] Soft Actor-Critic Algorithms and Applications, Haarnoja et al., 2019


### 2/ General motivation

The action-free setup is interesting, but it could be more motivated in this work. In particular, it would be valuable to have have more explanations -- or citations --  to support the claim that " [...] the action information in offline episodes can be difficult or even impossible to collect in some practical cases".

While I agree that a research question can be interesting by itself, I think it is important, if the work is motivated by a practical use case, to frame it properly and explain why this problem is important or useful.


### 3/ Framing of related work

I think this work lacks a presentation of the related literature of “learning from demonstrations” (LfD) i.e. the setup where one is given a dataset of demonstration and full access to the environment. This is different from imitation learning, where usually one does not assume access to the online return, and different from the current setup of the paper because of the presence of actions in the dataset. I still  think this work could to be framed within this existing literature, as adapting methods from LfD would make natural "naive" baselines for action-free RL. Some canonical methods include adding demonstrations in the replay buffer, as in  DDPGfD [1] and DQfD [2], or combining the actor critic with a BC loss as in [3].

I understand that the tackled setup here is different, since there is no action the dataset, but still I think would be an improvement to frame the work within the learning from demonstrations literature, which aims at the same goal, with just one more assumption on the data collection.

[1] Leveraging Demonstrations for Deep Reinforcement Learning on Robotics Problems with Sparse Rewards, Vecerik et al., 2017

[2] Deep q-learning from demonstrations, Hester et al., AAAI 2018.

[3] Learning Complex Dexterous Manipulation with Deep Reinforcement Learning and Demonstrations, Rajeswaran et al., 2018


### 4/ About reward design

Once an AFDT is trained, there seems to be plenty of possibilities for designing a guided reward. While the design of the reward makes sense, I think it lacks some motivation to explain why that particular choice was made. Specifically, it would be valuable to have a discussion around the optimal policy that this kind of reward induces. Here, it seems like an arbitrary choice.

Notably, since it is discussed why this reward does not lead to desirable behaviors, I think this discussion is even more important.


### 5/ Comparison to LfD

As mentioned above, I think it would be interesting to compare this method to an LfD approach (with actions in the dataset), to answer the questions “are the actions really needed in demonstrations?”. If the answer is negative, I would say this would be an intriguing result.

### 6/ Misc

Eq (2): it is not clear that a_{t+1} is sampled from \pi.

Figures: please state what the shaded areas represent.

---

### Note · Authors · 2024-03-27

I have read and agree with the venue's withdrawal policy on behalf of myself and my co-authors.